# The Efficacy and Safety of Deep Brain Stimulation of Combined Anterior Limb of Internal Capsule and Nucleus Accumbens (ALIC/NAcc-DBS) for Treatment-Refractory Obsessive-Compulsive Disorder: Protocol of a Multicenter, Randomized, and Double-Blinded Study

**DOI:** 10.3390/brainsci12070933

**Published:** 2022-07-16

**Authors:** Tingting Xu, Yuan Gao, Bin Li, Jiaxin Jiang, Huirong Guo, Xianzhi Liu, Hongxing Huang, Yuqi Cheng, Hualin Yu, Jie Hu, Xi Wu, Wei Wang, Zhen Wang

**Affiliations:** 1Shanghai Mental Health Center, Shanghai Jiao Tong University School of Medicine, Shanghai 200030, China; xutingting@smhc.org.cn; 2West China Hospital, Sichuan University, Chengdu 610041, China; gaoyuan_scu@foxmail.com (Y.G.); libinjia@163.com (B.L.); jjxftc2022@foxmail.com (J.J.); 3The First Affiliated Hospital of Zhengzhou University, Zhengzhou University, Zhengzhou 450052, China; drguohuirong@163.com (H.G.); fccliuxz@zzu.edu.cn (X.L.); 4Brains Hospital of Hunan Province, Changsha 410015, China; hhx541230@163.com; 5First Affiliated Hospital of Kunming Medical University, Kunming 650031, China; yuqicheng@126.com (Y.C.); yuhl308@126.com (H.Y.); 6Huashan Hospital, Fudan University, Shanghai 200040, China; 13361913896@163.com; 7Changhai Hospital, The Navy Military Medical University, Shanghai 200086, China; wuxi_smmu@sina.com

**Keywords:** obsessive-compulsive disorder, deep brain stimulation, anterior limb of internal capsule, nucleus accumbens

## Abstract

Backgrounds: Deep brain stimulation (DBS) is an emerging and promising therapeutic approach for treatment-refractory obsessive-compulsive disorder (OCD). The most common DBS targets include the anterior limb of internal capsule (ALIC) and nucleus accumbens (NAcc). This protocol aims to explore the efficacy and safety of the combined ALIC- and NAcc-DBS for treatment-refractory OCD. Methods: We will recruit 64 patients with refractory OCD from six centers, randomly allocate them to active and sham-stimulation groups through a three-month double-blind phase, then enter a three-month open-label phase. In the open-label stage, both groups experience real stimulation. Outcome measures: The primary outcome will be the efficacy and safety of combined ALIC- and NAcc-DBS, determined by treatment response rate between the active and sham-stimulation groups at the double-blind stage and spontaneously reported adverse events. The secondary outcomes are comparisons of change in Y–BOCS, CGI, HAMD, and HAMA scores at the third and sixth months compared to baseline between the active and sham-control groups, as well as the scores of the third month minus the sixth month between the two groups.

## 1. Introduction

Obsessive-compulsive disorder (OCD) is a common and disabling mental disorder that is characterized by intrusive thoughts or images (obsession) and repetitive, ritual behaviors (compulsion) induced by obsessions [1]. Its lifetime prevalence is about 2–3% worldwide [2,3]. Patients with OCD often suffer from severe anxiety and depression, and spend too much time on repetitive behaviors or inhibiting unwanted thoughts, making it difficult to finish their daily work and studies [4]. 

A preponderance of evidence supports the idea that the only established first-line treatments for OCD are exposure and response prevention (ERP), and selective serotonin reuptake inhibitor medications (SSRIs) [5,6]. Most patients show improvements in their symptoms with these interventions, either alone or in combination [7]. Beyond that, clomipramine and antipsychotic augmentation are optional approaches for OCD, with substantial empirical support [8,9,10]. However, even when a theoretically appropriate treatment is established, nearly 10–40% of patients do not obtain satisfactory amelioration of their clinical symptoms after a full dose and course of medication and psychotherapy, then develop into treatment-refractory OCD [11,12]. Thus, novel treatments, including repetitive transcranial magnetic stimulation (rTMS), transcranial direct current stimulation (tDCS), and even deep brain stimulation (DBS), are urgently needed. 

DBS is a well-accepted alternative to ablative surgery for movement disorders such as Parkinson’s disease and dystonia, and has been investigated for OCD. DBS is an intracranial, invasive, but reversible and controllable treatment, where electrical leads are implanted through stereotactic techniques into intracranial targets by minimal neurosurgery [13]. The DBS can release electrical pulses to suppress abnormal electroencephalograms, and reshape the neural functioning by reconstructing neural networks and neural transmitters [14]. 

DBS is a promising therapeutic approach for patients with treatment-refractory OCD. Based on the classical cortico-striato-thalamo-cortical (CSTC) circuit dysfunction model of OCD [15,16], the typical DBS targets implied in refractory OCD include the anterior limb of internal capsule (ALIC), ventral striatum (VS), the subthalamic nucleus (STN), and midbrain targets [17,18]. In 1999, Nuttin et al. [19] first reported that the DBS of ALIC (ALIC-DBS) was effective for refractory OCD. In this study, four patients received ALIC-DBS treatment; three of them showed a significant alleviation in symptoms. Among these, one patient’s symptoms were reduced by over 90% based on the parents’ self-report. The combined long-term results from four centers supported the idea that ALIC-DBS has long-term effects on OCD symptoms and is well-tolerated [20]. The VS target mainly refers to the nucleus accumbens (NAcc). NAcc-DBS has been associated with reduced OCD symptoms and decreased excessive frontostriatal connectivity between NAcc and the lateral and medial prefrontal cortex, partially supporting the hypothesis of extreme frontostriatal connectivity being involved in OCD pathophysiology [21]. Anteromedial subthalamic nucleus DBS (amSTN-DBS) is another therapeutic option for treating severe and refractory OCD, and has been found to be associated with significant improvements in social adjustment and struggles in work, social, and familial life [22]. The midbrain DBS location targeted ascending fibers of the ventral tegmental area (VTA), which travel through the superior limb of the medial forebrain bundle (slMFB), to connect with the superior frontal gyrus, the middle frontal gyrus, and the lateral orbitofrontal regions [23,24], which also suggests its treatment potential for OCD. In addition, the bed nucleus of the stria terminals (BNST) and the inferior thalamic peduncle (ITP) also have been used as potential targets in many clinical trials, with positive results [25,26]. A clinical review has reported a response rate of 58.2% and a mean symptom reduction of 47.7% across all DBS targets with treatment-refractory OCD [27]; however, most of the current clinical evidence comes from open-label cohorts or randomized controlled trials with very small sample sizes. Wu et al.’s review suggests that the DBS implanted in the ventral anterior internal capsule represents an emerging, but not established, therapy; additional, well-designed, and blinded clinical trials are still needed [28]. Based on the aforementioned CSTC circuits dysfunction model, the VS, NAcc, and nearby ALIC have become the most common targets; however, the DBS implantation of NAcc or ALIC alone is currently not enough because it does not address the impairments in complex emotional and cognitive processing, as well as OCD symptoms. Combined multiple targets of DBS could be a novel direction for treatment-refractory OCD. 

Huys et al. [29] investigated the DBS of the combined anterior limb of internal capsule and nucleus accumbens (ALIC/NAcc-DBS) for refractory OCD for the first time. They reported a 40% response rate and a 33% reduction in Y–BOCS, indicating that ALIC/NAcc-DBS is a promising treatment option. However, this study was based on an open-label and small sample cohort, and the two targets were stimulated under the same parameters, which cannot effectively activate the two targets simultaneously. Previous studies have shown that the NAcc and ALIC are gray-matter neurons and white-matter fiber bundles, respectively, which require different parameters to show clinical benefits. A review of earlier clinical trials showed that the parameters used in ALIC were higher than those used in NAcc, especially voltage and pulse width [25,30,31,32,33]. 

The present study aims to explore the efficacy and safety of ALIC/NAcc-DBS for treatment-refractory OCD, based on a multicentered, randomized, double-blinded, and sham-controlled design. The DBS system used in this clinical trial adopts a multi-source stimulation mode, which means that different stimulation parameters can be used in NAcc and ALIC to simultaneously meet the effective stimulation requirements of the two neural structures. Custom tetrapolar electrodes will be used to achieve broad coverage of ALIC and NAcc. 

## 2. Materials and Methods

### 2.1. Study Description and Design

This study aimed to evaluate the treatment effects and tolerability of bilateral ALIC/NAcc-DBS for refractory OCD. This was achieved through a multicenter, randomized, double-blind, and active-sham control clinical trial, which was performed at six centers in China. The dominant centers are Shanghai Mental Health Center and West China Hospital affiliated to Sichuan University; the others include the Huashan Hospital affiliated with Fu Dan University, The First Affiliated Hospital of Zhengzhou University, The First Affiliated Hospital of Kunming Medical University, Hunan Brain Hospital, and The First Affiliated Hospital of Navy Medical University. The Ethics Committee has approved the study at each center. 

### 2.2. Patient Selection

Sixty-four patients with OCD who met the eligibility criteria were recruited from six centers. All patients voluntarily came to our institution and independently chose to receive the surgery. A signed informed consent form was obtained from each patient prior to the study. 

#### 2.2.1. Inclusion Criteria

The patients were eligible for recruitment if they meet the following criteria: (1) age range from 18 to 65 years old; (2) met the Diagnostic and Statistical Manual of Mental Disorders, fifth edition (DSM-5) criteria for OCD as the primary diagnosis; (3) had a score of at least 25 on the Yale–Brown Obsessive-Compulsive Scale (Y–BOCS), accompanied by substantial functional impairment; (4) met treatment-refractory criteria. Treatment refractory is defined as no response to: (a) a minimum of three adequate pharmacological trials with first- and second-line medications (at least one trial using clomipramine), with doses at or tolerated beyond the maximum recommended dose; (b) augmentation with at least two antipsychotic drugs; (c) adequate trials of cognitive-behavioral therapy (defined as a minimum of 20 sessions of therapist-guided exposure and prevention therapy).

#### 2.2.2. Exclusion Criteria 

Patients will be excluded if they meet any of the following criteria: (1) comorbid schizophrenia spectrum and other psychotic disorders, bipolar disorder, major depression disorder, etc.; (2) severe neurological or physical illness; (3) contraindications to neurosurgery; (4) substance abuse or dependence; (5) pregnant females; (6) severe suicide risk and tendencies. 

### 2.3. Procedures 

According to the inclusion criteria, the participants must attest to maintaining stable pharmacological treatment for two months preceding DBS surgery and throughout participation in the DBS trial until the end of the follow-up. The visit time points include a screening visit, preoperative baseline visit, perioperative period, programming/stimulation adjustment, double-blind visit, open-label visit, and extended one-year observation. The flow chart is shown in Figure 1. 

During the screening visit, patients who meet the criteria for refractory OCD are screened after signing informed consent. At this stage, patients undergo a series of laboratory and imaging examinations to exclude potential contraindications for surgery or DBS treatment, including blood routine, urine routine, alanine aminotransferase (ALT), aspartate aminotransferase (AST), creatinine, urea nitrogen, blood coagulation, blood pregnancy (exclusively for women of childbearing age), electrocardiogram, chest X-rays, cerebral magnetic resonance imaging (MRI) or head CT, etc. 

The preoperative baseline visit will take place two weeks before the DBS operation. Patients will be screened according to existing inclusion/exclusion criteria, then undergo the clinical interview and a battery of clinical assessments. Demographic information (including age, gender, education, etc.), past and current medication status, and other clinical information will be recorded at baseline. The Y–BOCS and Y–BOCS symptom checklist will be used to assess the severity of OCD symptoms and list all the OCD symptoms. The Mini-International Neuropsychiatric Interview (MINI) will be used to screen comorbid DSM-5 psychiatric disorders. The Columbia suicide severity rating scale (C−SSRS) will be used to assess suicidal ideas or behaviors in participants. The Hamilton anxiety scale (HAMA), Hamilton depression scale (HAMD), and clinical global impressions (CGI) scale will be used to measure the severity of clinical symptoms. The Chinese versions of all measures have been shown to be reliable and valid. 

After surgery, a 1.5 T MRI will be conducted in all participants to ensure that the electrodes are in the targeted position and the lead placement is accurate. Then, all patients will be randomized and assigned to the active-stimulation group (experimental group), and the sham-stimulation group (control group) by a randomized allocation system (SceneRay, Suzhou, China), and then integrated into the programmer according to the randomization plan. About 2–4 weeks after surgery, patients will undergo the first programming to adjust the stimulus parameters when recovering from cerebral edema. The parameters are optimized and individualized based on the acute effects on mood, anxiety, or self-reported OCD symptoms. During the exploration, subjects should not actively be exposed to an obsessive-compulsive symptom-inducing situation to assess their response to stimulation parameters. The active and sham-stimulation groups will be evaluated in the same way. During the study, the parameters will remain unchanged, unless for medical reasons. 

After the initial parameter optimization phase, patients enter a three-month, double-blind follow-up process, during which the active group is stimulated (ON phase) while the sham group is not (OFF phase). After that, the allocation is uncovered, patients enter a three-month, open-label, follow-up process, the sham group switch to a true stimulation of DBS, and the experimental group maintains the original stimulation. At the end of the double-blind and open-label processes, patients are evaluated using standardized psychiatric questionnaires (Y–BOCS, CGI, HAMD, HAMD, and C-SSRS). In addition, each patient’s medication will be recorded and maintained at a stable dose, so that we can investigate the effect of stimulation as a single therapy. Patient recruitment and screening are currently underway.

### 2.4. Blinded Design 

This protocol is designed for three blinded parties; that is, participants, programmers, and assessors. The programmers adjust the stimulus parameters through a special program software, which will preset the random coding program to group the subjects. The software interface is displayed as real in both groups, while the actual stimulation output is distinguished by the software, so that the programmers cannot distinguish between the subject groups. The outcome assessor should be a professional psychiatric doctor who is independent of the surgery and program periods and does not know the subject grouping. He/she will complete the Y–BOCS and CGI outcome measures for the baseline and follow-up periods.

### 2.5. Surgery

All centers (except for clinical research institutions that do not assume surgical responsibilities, i.e., Shanghai Mental Health Center) have the expertise of neurosurgeons with more than five years of experience performing DBS surgery. The DBS surgery will follow a standard operating procedure for perioperative management and stereotactic procedures. DBS electrodes’ placement will be decided by a preoperative MRI scan and postoperative image using a Leksell surgical planning system (SurgiplanTM, Elekta, Sweden). Custom tetrapolar electrodes (1242, SceneRay) will be inserted along the trajectory of the ALIC [34,35], extending into NAcc, with lead contacts of 3.0 mm long and contact spacing of 2.0 mm, 4.0 mm, and 4.0 mm, from ventral to dorsal, respectively. Electrode leads will be externalized to confirm the electrode locations and perform a temporary stimulation. The NAcc targets (contact 0) will be set 7–12 mm lateral to the midline, 5–7 mm anterior to the anterior border of the anterior commissure, and 4–6 mm inferior to the inter-commissural line for reference [36]. The two ventral contacts are preset with the NAcc, while the two dorsal contacts are preset into the ALIC. A connecting wire (SR1341, SceneRay) from the scalp connects to a subcutaneous implantable pulse generator (IPG; SR1181, SceneRay). The IPG, with a non-rechargeable battery, will be subcutaneously implanted at the right subclavicular area. After implantation, patients will be monitored overnight to prevent possible complications, including hemorrhage or infection. A head CT will be obtained within 24 h to screen for intracranial hemorrhage. If significant complications are absent, the patient will be discharged from 5 to 7 days after the surgery with OFF-stimulation to resolve cellular reactions to insertion. 

### 2.6. Adverse Events 

All adverse events (AEs) and device-related AEs will be documented throughout the study and then included in the final analysis. The records will consist of the names of the AEs, time of occurrence, severity, severe adverse events (SAEs), the relationship with the device, corresponding treatment measures, and the outcomes of AEs. SAE is defined as any situation that could result in death, life-threatening events, or significant deteriorations in health during the experiment, including fatal illness or injury, hospitalization or prolongations in existing hospitalization, significant disability/incapacity, or intervention to prevent permanent impairment. The possible AEs include postoperative pain, stress, or discomfort; intracranial hemorrhage; subcutaneous hemorrhage or seroma; infection; seizure or convulsions; aphasia; cranial neuropathy; amnesia; paralysis; death; cerebrospinal fluid leakage; etc. [37]. Device-related AEs mean adverse events induced by DBS device defects. The device defects that occur during the study will be documented, including the name and identification number of the device, the cause, the time of occurrence, the relevance to AE or SAE, the outcome, and a detailed description of the device defects. The participants will be informed that implantation of the DBS system involves the above risks before signing the informed consent. When AEs occur, the researchers should offer the appropriate treatment and follow-up according to the specific AE situation until the symptoms disappear or stabilize. 

### 2.7. Statistics

#### 2.7.1. Sample Size

The study is a randomized, double-blinded, active-sham controlled design, the statistical experts decided on a 1:1 sample ratio for the experimental group to the control group. Based on the previous literature reports, the treatment response rate of different DBS targets for refractory OCD should range from approximately 40% to 60% [38], and previous studies showed that the remission rate of patients with OCD who do not receive any treatment is around 10% [39]. Therefore, the estimation of the overall efficacy of the experimental group is set to the median of 50%; the estimation of the overall effectiveness of the control group is set at 10%. This study is designed as a differential test to determine whether the mean of the experimental group (u1) is different from that of the control group (u2). The hypotheses are as follows: H0: u1 − u2 = 0, H1: u1 − u2 ≠ 0. A two-sided significance level was set at 5% and statistical power at 90%, with a maximum acceptable drop rate of 20%. The total sample size of the study was 64, and the size of each group was 32. The sample size was calculated by the PASS V.11 sample size calculation software (NCSS, Kaysville, UT, USA). 

#### 2.7.2. Data Management

The electronic data collection (EDC) system will be used to collect all experimental data. The EDC system was rigorously tested and is fully compliant with the criteria of the Code of Quality Management for Clinical Trials of Medical Devices and the Technical Guide to Data Management in Clinical Trials. System testing and training of relevant personnel will be conducted before the EDC is officially enabled. When formally enabled, the researchers will be provided with an account number and password. The account is related to the user’s responsibilities and permissions. The relevant personnel shall properly hold the account information and shall not disclose the account information to others.

#### 2.7.3. Data Analysis

We will prepare the statistical analysis plan (SAP) based on the research program and database, applying the Statistical Analysis Software (SAS) version 9.4. Descriptive statistics use standard deviations or medians with interquartile ranges for continuous variables (depending on data distributions) and proportions for categorical variables. The comparison of group differences will be conducted using appropriate differentiate testing depending on the data type, including paired or group *t*-tests, analysis of variance (ANOVA), Wilcoxon rank-sum test, Kruskal–Wallis H rank-sum test, Chi-square test, Cochran Mantel Haenszel (CMH), etc. The significant level α will be set as 0.05 with the two-tailed test, and statistical uncertainty will be expressed in 95% confidence intervals (CI). 

## 3. Outcomes

### 3.1. Primary Outcome

The primary outcome is a comparison of the treatment response rate (TRR) of the active and sham-controlled groups at the third month after initial stimulation. (TRR is defined as the number of treatment responses/the number of groups, a ≥35% reduction in Y–BOCS plus CGI ≤ 2, meeting the treatment response criteria.) The safety evaluation is determined by spontaneous reports, and laboratory testing is reflected in AEs. As mentioned in the Methods section, all AEs will be documented throughout the whole study.

### 3.2. Secondary Outcomes

The secondary outcomes are comparisons of change in Y–BOCS, CGI, HAMD, and HAMA scores at the third and sixth months compared to baseline between the active and sham-control groups, as well as the scores of the third month minus the sixth month between the two groups. 

#### 3.2.1. Yale–Brown Obsessive-Compulsive Scale 

The Y–BOCS is a 10-item, clinician-rated questionnaire, widely used to assess the severity of OCD symptoms, and sensitive to measuring treatment effect. This has been shown to have good reliability and validity [40]. A higher score means a higher symptom severity of OCD.

#### 3.2.2. Clinical Global Impressions 

The CGI is a clinician-rated scale to assess treatment response in patients with mental disorders, which is used by clinicians to rate the extent to which the patient’s illness has improved or worsened relative to baseline measurements [41]. 

#### 3.2.3. Hamilton Depression Rating Scale 

The HAMD-24 is a multiple-item questionnaire, which is widely used by clinicians to quantify the severity of depression symptoms, with a higher score meaning a higher severity of depression [42]. 

#### 3.2.4. Hamilton Anxiety Rating Scale 

The HAMA is a 14-item questionnaire used by clinicians to assess the severity of anxiety symptoms, with a higher score indicating a higher severity of anxiety [43]. 

## 4. Discussion

To the best of our knowledge, this is the first multicentered, double-blinded, sham-controlled study protocol to evaluate the therapeutic efficacy and safety of bilateral ALIC/NAcc-DBS for patients with treatment-refractory OCD. The sample size of this study protocol was relatively large; to the best of our knowledge, it is the largest sample size in a single study worldwide at present. The size of the Chinese population ensured that we could recruit enough eligible patients. 

The main concerns are the risks of any DBS neurosurgical procedure and long-term neuro-modulation. The neurosurgical-related side effects mainly include postoperative pain or discomfort, intracranial hemorrhage, infection, etc. These are mainly avoided by surgical operation. In this study, the surgeons will have extensive experience in neurosurgery and experience in DBS implant surgery. On the other hand, the risks related to long-term neuro-modulation mainly include new illness or injury, worsening or acceleration in previous condition, insomnia, hypomania, mania, and most seriously, seizures. According to previous studies, seizures emerged after 2–5 years of stimulation. To better observe the long-term adverse effects of neuro-modulation, we designed a one-year extended observation after the open-label trials.

In conclusion, this study is a multicentered, double-blinded, sham-stimulation-controlled study to investigate the efficacy and safety of ALIC/NAcc-DBS for refractory OCD. We hope that our study will benefit the patients who participate. We also hope that this study could extend DBS target options in future studies, to help patients who suffer from treatment-refractory OCD. 

## Figures and Tables

**Figure 1 brainsci-12-00933-f001:**
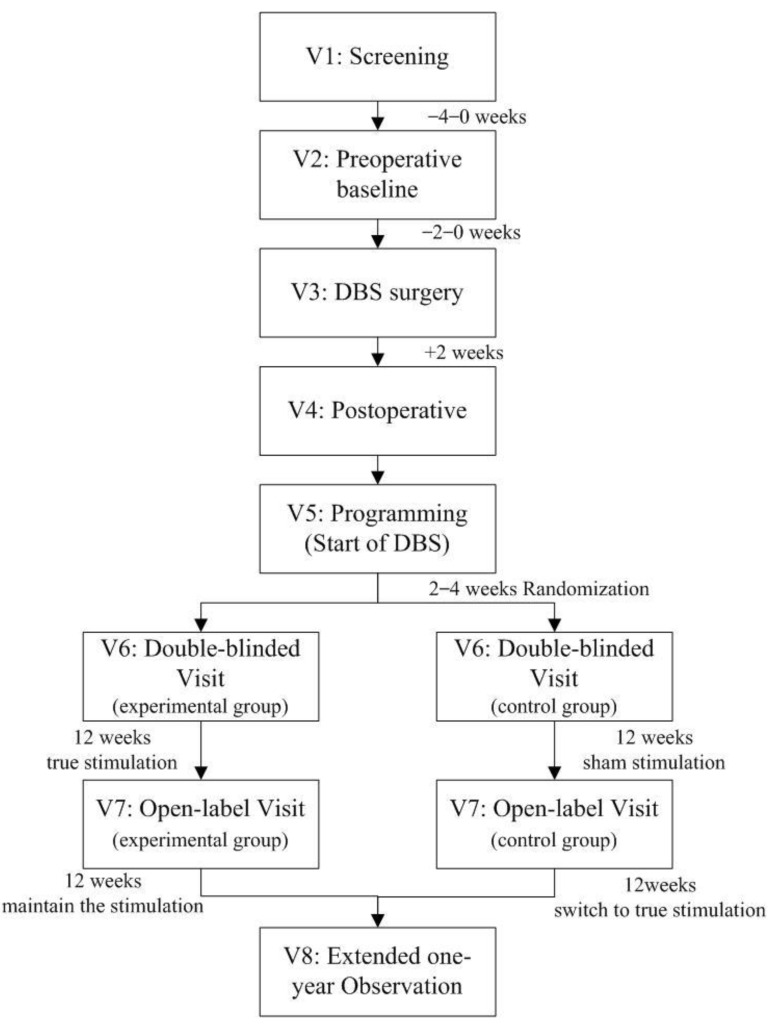
Flow chart of the study.

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
