# Peer review of "The Efficacy and Safety of Deep Brain Stimulation of Combined Anterior Limb of Internal Capsule and Nucleus Accumbens (ALIC/NAcc-DBS) for Treatment-Refractory Obsessive-Compulsive Disorder: Protocol of a Multicenter, Randomized, and Double-Blinded Study"

_brainsci, 2022, doi:10.3390/brainsci12070933_

Round 1

Reviewer 1 Report

The authors present a study protocol addressing the relevant and important issue of DBS therapy for treatment refractory OCD with ALIC/NAcc as target. They outline a well structured multicenter double-blinded clinical trial.

I would suggest the following minor changes:

1.       Line 61-63: In the Introduction the authors state several targets – however – omitting the superolateral Medial Forebrain Bundle (slMFB or VTA projection pathway = VTApp) first described by Coenen et al. and then also found by Horn and colleagues, albeit named differently by them. The slMFB/VTApp could be the common target structure of all targets named by the authors. As the issue of connectivity is of increasing importance in functional neurosurgery, it should be addressed in the introduction in more depth.

2.       Line 123: To my knowledge for diagnosis of OCD it is a prerequisite, that symptoms are not better explained by the symptoms of another mental disorder, among them psychotic disorders. So this sentence does not make sense to me.

3.       Lines 187 onwards: The implantation of the DBS system should be described in more detail, e.g. are tractographic methods used in addition?, the target coordinates reported by the authors differ a lot from those reported by the cited work of Huff et al. Did the authors rely on further publications to derive the coordinates? I guess the implant is non-rechargeable, but please state clearly (and not only giving a product code), whether the implanted device is rechargeable or not, as it is relevant for blinding procedures.

4.       Line 204-205: This sentence is contradictory to Line 159.

5.       A general point to be addressed throughout the manuscript is proper English language editing.

Reviewer 2 Report

This piece of research study aims to examinr the efficacy and safety of ALIC/NAcc-DBS for 89 treatment-refractory OCD, based on a multi-centered, randomized, double-blinded, and 90 sham-controlled design. This is an interesting manuscript and I only have minor points:

-What were the psychometric properties of the Y-BOCS and is it validated for the study population? The authors of the adaptation and the properties, if possible, for this sample should be cited.

-Figure 1 is blurred

-There is not need to justify sample with a formula, but other question would be of interest to review - is there sufficient statistical power?

Reviewer 3 Report

There are two issues with this paper – one has to do with the protocol and another with the language.

In terms of protocol, three months of randomization will not be enough to see any meaningful difference; this was the reason for failure of the earlier DBS study to show efficacy of stimulation of subcallosal cingulum in depression (BROADEN) as the sham cohort was very similar to active treatment and the difference became obvious only after 6 months from baseline.

Second problem is the language: throughout the paper the research is described in past, present and future tenses – this has to be changed so the language is uniform.

In addition, there are multiple instances of misuse of the words: “contradiction to surgery” (twice), “preparing for pregnant”, “chart of the study shows in Figure”, “designed to blinded to three party”, “protocol to evaluating”

Round 2

Reviewer 3 Report

looks much better

Author Response

On behalf of my co-authors, we thank you very much for giving us an opportunity to revise our manuscript, and we also appreciate editors and reviewers very much for your positive and constructive comments and suggestions on our manuscript.
